∂ | **Open Peer Review** | Human Microbiome | Research Article

# Species-level verification of *Phascolarctobacterium* association with colorectal cancer

Cecilie Bucher-Johannessen,[1,2,3] Thulasika Senthakumaran,[4] Ekaterina Avershina,[2,3] Einar Birkeland,[5] Geir Hoff,[6,7] Vahid Bemanian,[8] Hege Tunsjø,[4] Trine B. Rounge[1,2,3]

**ABSTRACT** We have previously demonstrated an association between increased abundance of *Phascolarctobacterium* and colorectal cancer (CRC) and adenomas in two independent Norwegian cohorts. Here we seek to verify our previous findings using new cohorts and methods. In addition, we characterize lifestyle and sex specificity, the functional potential of the *Phascolarctobacterium* species, and their interaction with other microbial species. We analyze *Phascolarctobacterium* with 16S rRNA sequencing, shotgun metagenome sequencing, and species-specific qPCR, using 2350 samples from three Norwegian cohorts—CRCAhus, NORCCAP, and CRCbiome—and a large publicly available data set, curatedMetagenomicData. Using metagenome-assembled genomes from the CRCbiome study, we explore the genomic characteristics and functional potential of the *Phascolarctobacterium* pangenome. Three species of *Phascolarctobacterium* associated with adenoma/CRC were consistently detected by qPCR and sequencing. Positive associations with adenomas/CRC were verified for *Phascolarctobacterium succinatutens* and negative associations were shown for *Phascolarctobacterium faecium* and adenoma in curatedMetagenomicData. Men show a higher prevalence of *P. succinatutens* across cohorts. Co-occurrence among *Phascolarctobacterium* species was low (<6%). Each of the three species shows distinct microbial composition and forms distinct correlation networks with other bacterial taxa, although *Dialister invisus* was negatively correlated to all investigated *Phascolarctobacterium* species. Pangenome analyses showed *P. succinatutens* to be enriched for genes related to porphyrin metabolism and degradation of complex carbohydrates, whereas glycoside hydrolase enzyme 3 was specific to *P. faecium*.

**IMPORTANCE** Until now *Phascolarctobacterium* has been going under the radar as a CRC-associated genus despite having been noted, but overseen, as such for over a decade. We found not just one, but two species of *Phascolarctobacterium* to be associated with CRC—*Phascolarctobacterium succinatutens* was more abundant in adenoma/CRC, while *Phascolarctobacterium faecium* was less abundant in adenoma. Each of them represents distinct communities, constituted by specific microbial partners and metabolic capacities—and they rarely occur together in the same patients. We have verified that *P. succinatutens* is increased in adenoma and CRC and this species should be recognized among the most important CRC-associated bacteria.

**KEYWORDS** *Phascolarctobacterium*, colorectal cancer, microbiome

Address correspondence to Trine B. Rounge, t.b.rounge@farmasi.uio.no.

The authors declare no conflict of interest.

See the funding table on p. 17.

Many studies have revealed associations between the microbiome and several intestinal diseases. Among others, imbalance in microbial composition and enrichment of specific intestinal bacteria have been associated with adenoma formation and their subsequent progression to CRC *via* the adenoma-carcinoma pathway (1). The time span for the progression can vary from 5 to 10 years depending on the specific

pathway of tumorigenesis (2). However, less than 10% of the adenomas are estimated to progress to cancer (3, 4).

We have previously shown an increased abundance of Amplicon Sequence Variants (ASV) belonging to the genus *Phascolarctobacterium* in CRC and adenoma cases when compared to healthy controls in stool samples and tissue samples in two independent Norwegian cohorts (5, 6).

Three species of *Phascolarctobacterium, Phascolarctobacterium succinatutens, Phascolarctobacterium faecium,* and *Phascolarctobacterium wakonense,* have been described previously but they remain largely uncharacterized. While *P. wakonense* has been isolated from common marmoset feces (7), *P. succinatutens* and *P. faecium* are abundant in the human gastrointestinal (GI) tract (8, 9). *P. succinatutens* is estimated to be present in around 20% of human fecal samples while the prevalence of *P. faecium* varies between 40% and 90%, being strongly influenced by host age (8). The genus is a Gram-negative, obligate anaerobic bacteria belonging to Negativicutes class in the phylum Firmicutes. Both *P. faecium* and *P. succinatutens* use succinate as an energy source and can convert succinate into propionate (9, 10). However, they lack the fumarate reductase gene, an enzyme essential for the conversion of fumarate into succinate (11); thus, they rely on the presence of succinate from the environment. Succinate, a tricarboxylic acid (TCA) cycle intermediate in humans, is not abundant in the human diet but is produced in the GI tract by the host and bacteria such as those belonging to *Paraprevotella* (9) and *Bacteroides* (10).

Studies have reported an association between *Phascolarctobacterium* and adenoma/CRC. Yachida et al. (12) observed an enrichment of *P. succinatutens* in early CRC stages, accompanied by elevated succinate levels. Also, Zackular et al. (13) and Peters et al. (14) have reported a higher abundance of *Phascolarctobacterium* in fecal samples from adenoma/CRC cases. By contrast, a small study by Sarhadi et al. (15) found a reduced abundance of *Phascolarctobacterium* in fecal samples from CRC compared to controls. While these studies showed an association between adenoma/CRC and *Phascolarctobacterium* along with several other bacteria, none of them conducted in-depth analyses on the species level.

We aimed to verify the association between *Phascolarctobacterium* and adenoma/CRC at the species level using independent cohorts and techniques and to compare the genomic makeup of *Phascolarctobacterium* across species.

## MATERIALS AND METHODS

### Study population and sample collection

Data from the CRC study from Akershus University Hospital (CRCAhus hereafter), the Norwegian Colorectal Cancer Prevention (NORCCAP) trial and CRCbiome study, and a publicly available data set, curatedMetagenomicData, were included in this study (Fig. 1).

The CRCAhus study (for details, see Senthakumaran et al. (6)) includes 72 participants (age 30–87) who underwent colonoscopy at Akershus University Hospital between 2014 and 2017. Individuals included in the study were either referred for colonoscopy following the detection of polyps by computed tomography or undergoing investigation for CRC due to unexplained bleeding or altered stool patterns for more than 4 weeks. Based on colonoscopy findings, the participants were classified into three categories: patients with cancer, patients with adenomatous polyps (diameter ≥10 mm), and healthy controls (no pathological findings). Either two or four biopsy samples from different locations in the colon were collected during colonoscopy for controls and cases, respectively. Each participant collected a stool sample in the RNALater RNA stabilizing buffer (Thermo Fisher Scientific, Waltham, MA, USA) before colonoscopy or 1 week after colonoscopy. In total, the study population included 72 participants. Of these, 70 (CRC = 23, adenoma = 25, controls = 22) provided stool samples, and 60 biopsy samples were included (one from each participant; CRC = 23, adenoma = 20, controls = 17). Detailed information on the participants and sample collection was described elsewhere (16).

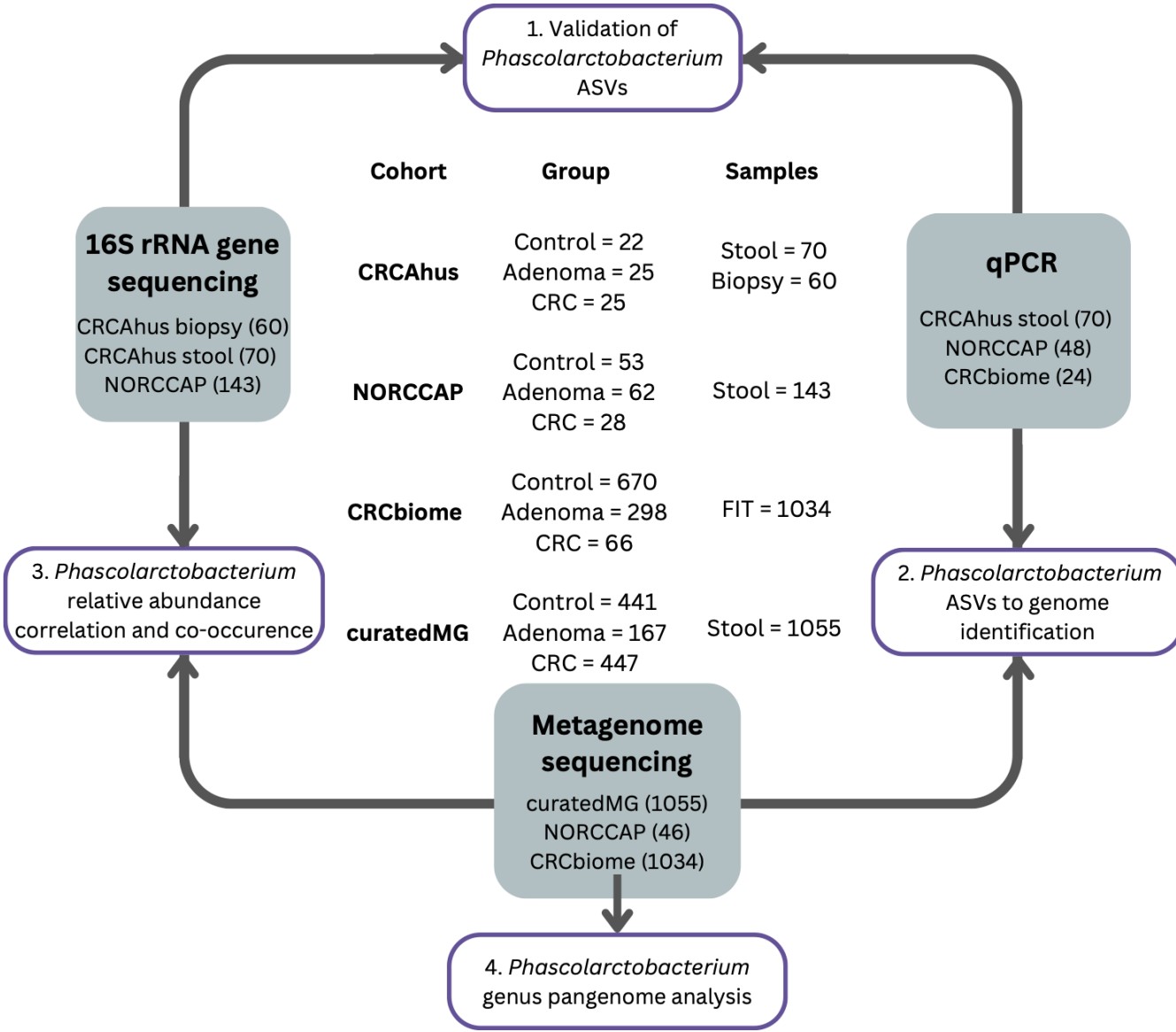

**FIG 1** Cohorts, processing workflow, and data analyses included in this study. ASV = Amplicon sequence variant; FIT = fecal immunochemical test.

The NORCCAP trial (for details, see Holme et al. (17, 18) and Bretthauer et al. (19)), took place in 1999–2001 and recruited participants (age 50–65) from the Norwegian counties of Oslo and Telemark. Participants collected fecal samples at home in 20 mL vials and immediately stored them in their freezers for up to 7 days, until transportation to the screening center during their sigmoidoscopy screening appointments, and further storage at −20°C. In all, 28 participants were diagnosed with CRC at screening or diagnosed up to 17 years after screening (identified through cancer registry linkage in 2015). In total, 63 participants had high-risk adenomas classified at the time of sigmoido-scopy screening. Finally, 53 participants were included as healthy controls (no adenoma or cancer diagnosis at screening or cancer during follow-up). Participants with high-risk adenomas were defined as having one or more adenomas ≥10 mm, with high-grade dysplasia or villous components regardless of size; or having three or more adenomas regardless of their size, dysplasia, and villosity.

The CRCbiome study (for details see Kværner et al. (20)) recruited participants from the Bowel Cancer Screening in Norway (BCSN) trial (21) between 2017 and 2021. Participants in the BCSN trial were invited for once only sigmoidoscopy or biennial

fecal immunochemical test (FIT). CRCbiome recruited participants (age 50–74) from the FIT arm, inviting those with a positive FIT test (>15 µg hemoglobin/g feces) who were referred for colonoscopy. Based on diagnoses retrieved from the BCSN database, participants were divided into three groups including 66 CRC cases, 298 advanced adenomas (including advanced adenomas, and advanced serrated lesions), and 670 controls (including no findings and those with non-advanced adenomas <3 mm). The CRCbiome study aims to explore the influence of diet and lifestyle on the microbiome. Participants completed two questionnaires prior to colonoscopy: A Food Frequency Questionnaire (FFQ), encompassing 256 food items across 23 questions about consumption frequency, portion sizes, and BMI; and a Lifestyle and Demographic Questionnaire (LDQ) with 10 items (20, 22). From this, a healthy lifestyle index (HLI) was developed as described in Kværner et al. 2023 (23).

For external validation, we utilized the publicly available R package (data set) CuratedMetagenomicData (24) (accessed 22.03.2022, curatedMG hereafter), a comprehensive data set of 22,588 samples obtained from 93 independent data sets. Samples are collected from various body sites, and raw data are processed to generate relative abundance tables using MetaPhlAn3. We filtered the data to only include samples from stool and conditions including CRC, adenomas, and healthy controls. This resulted in a subset of 1,055 samples from seven different studies where 447 were from CRC, 147 from adenomas, and 441 from healthy controls. The largest study included in this data set was from Yachida et al. (12) with 576 samples.

## DNA extraction and sequencing

DNA from biopsies and fecal samples from CRCAhus were extracted using AllPrep DNA/RNA Mini Kit (Qiagen, Hilden, Germany) and PSP Spin Stool DNA Kit (Stratec Molecular Gmbh, Berlin, Germany), respectively, as described in reference (16). Amplicon sequencing of 16S rRNA V4 region was performed on the Illumina MiSeq platform (Illumina Inc., San Diego, CA, USA) using the MiSeq reagent kit v/2 as previously described (6). PCR amplification of 16S rRNA V4 region was performed using 16S forward primer (16Sf V4: GTGCCAGCMGCCGCGGTAA) and 16S reverse primers (16Sr V4: GGACTACHVGGGTWTCTAAT) (25).

DNA extraction of the NORCCAP samples was performed using the QIAsymphony automated extraction system and a QIAsymphony DSP Virus/Pathogen Midi Kit (Qiagen, Hilden, Germany), with an off-board lysis protocol that included modifications. The process involved bead beating of the samples, followed by a mixture with a lysis buffer, and subsequent incubation for lysis. Amplification of 143 samples was carried out using a TruSeq (TS)-tailed one-step amplification protocol (26). For 16S rRNA sequencing, the V3-V4 region was targeted using the primers S-D-Bact-0341-b-S-17 (5′CCTACGGGNGGCWGCAG′3) and SD-Bact-0785-a-A-21 (5′GACTACHVGGGTATC-TAATCC′3) (27). Sequencing was performed using the Illumina MiSeq instrument generating paired-end reads of 2 × 300 bp. A subset of the samples, 46, was also metagenome sequenced using the Riptide protocol (Twist Bioscience, CA, USA) and sequenced on an Illumina NovaSeq platform, generating paired-end reads of 2 × 130 bp. We have previously shown the feasibility of using these long-term stored samples for microbiome analyses (28).

For CRCbiome, DNA extraction followed a similar protocol as NORCCAP, but with the inclusion of an extra washing step during lysis. The sequencing libraries for 1034 CRCbiome samples were prepared in line with the Nextera DNA Flex Library Prep Reference Guide with the modification of reducing the reaction volumes to a quarter of the recommended amounts. Sequencing was performed on the Illumina Novaseq system generating 2 × 151 bp paired-end reads (Illumina, Inc., CA, USA).

## Bioinformatics and taxonomic profiling

16S rRNA sequencing data from CRCAhus and NORCCAP were processed using Quantitative Insights Into Microbial Ecology (QIIME2 (29), version 2021.2.0 and 2020.2.0,

respectively) with the DADA2-plugin as described previously (5, 6), resulting in ASV. For comparative analysis with metagenomic data, ASV counts were transformed to relative abundances using the transform_sample_counts function from the Phyloseq package(30) (v1.26.1) where each ASV count was divided by the total count of ASVs in the sample.

NORCCAP metagenomic reads were processed using Trimmomatic (31) (v0.66.0) for quality trimming, discarding sequences below a quality threshold of 30 across four bases and those shorter than 30 base pairs. Bowtie2 (32) (v2.4.2) and Samtools (33) (v1.12) were used for the removal of human-derived sequences. Taxonomic profiling was conducted using MetaPhlAn3(34) (v3.0.4) with default settings.

For the CRCbiome samples, sequencing reads were processed using two different approaches. First, raw reads were trimmed using Trimmomatic (v0.36), and read mapping to the human genome (hg38) and PhiX were removed using Bowtie2 (v2.3.5.1). Read-based taxonomy was determined at the species level and quantified as relative abundance determined by MetaPhlAn3 using the mpa_v30_Chocophlan_201901 (v3.0.7) database. Second, metagenome-assembled genomes (MAGs) were created using the framework Metagenome-ATLAS (35) (v2.4.3). Low-quality reads were filtered and human and phiX sequences were removed using BBTools (36). Reads were then assembled *via* MetaSpades (37) (v3.13) and grouped into genomes with DAStool (38) (v1.1), utilizing MetaBat (39) (v2.2) and MaxBin (40) (v2.14) for genomic bin identification. Genome dereplication was conducted using dRep (41) (v2.2) based on 95% identity over 60% genome overlap. Genomes with completeness >90% and contamination <10%, determined using CheckM (42) were kept. GTDB-Tk (v1.3) assigned a taxonomy against the GTDB database (43) (v95). Metagenome-assembled genome abundance was estimated by median read depth across 1,000 bp bins of each genome and scaled by reads per million. The taxonomic classification approach was employed for those analyses where consistency and comparability across data sets were necessary. MAGs were used for analyses including CRCbiome samples which encompassed genomic characterization and functional potential of the individual *Phascolarctobacterium* species.

## Species-specific quantification of *Phascolarctobacterium* by qPCR

To verify our previous findings between *Phascolarctobacterium* ASVs and adenoma/CRC (5, 6), we developed species-specific qPCR assays. BLAST (Basic Local Alignment Search Tool) search identified the ASVs as *P. succinatutens* and *Phascolarctobacterium* sp. 377. As *P. faecium* is also prevalent in the human GI tract, we decided to include this species as well, and genomes from *P. succinatutens* (YIT 12067), *P. faecium* (JCN 30894), and *P.* sp 377 (AB739694.1) were used for qPCR assay development. IDT PrimerQuest Tool (Integrated DNA Technologies, Leuven, Belgium) was used for primer and probe design. The primers and the probes were synthesized by TiB Molbiol (Berlin, Germany) and are listed in Table 1. The analytical specificity of the *Phascolarctobacterium* qPCR assays was tested using 50 different bacterial strains, obtained mostly from the Culture Collection University of Gothenberg (CCUG) and clinical isolates from Akershus University Hospital (Table S1). The limit of detection (LOD) was determined using a 10-fold serial dilution of DNA from pure bacterial suspensions. qPCR assays were performed using Brilliant III Ultra-fast QPCR master mix (Agilent, Santa Clara, CA, USA) with 2 µL DNA in 20 µL reaction volume. Amplification was conducted on QuantStudio5 Real-Time PCR systems (Thermo Fisher Scientific). Cycling conditions for the *Phascolarctobacterium* assays were as follows: an initial denaturation of 95℃ for 5 min, followed by 40 cycles of 95℃ for 15 s, 60℃ for 30 s, and 72℃ for 30 s.

In the CRCAhus cohort, all 70 fecal samples were subjected to species-specific qPCR analysis. For the NORCCAP cohort, 38 samples with reads mapping to the genus level of *Phascolarctobacterium* in both the 16S rRNA and metagenome data sets were subjected to species-specific qPCR, along with 10 samples with no reads mapping to *Phascolarctobacterium*. In addition, 24 samples from the CRCbiome cohort were selected to verify the detection of *Phascolarctobacterium* and identify their genomes. The CRCbiome cohort

**TABLE 1**  List of primers and TaqMan probes used in this study

| | Target | Primer and probe sequence 5'-3' | Amplicon size | Reference |
|---|---|---|---|---|
| *P. succinatutens* | 16S rRNA | Fwd: GGGACAACATCCCGAAAGG | 73 | This study |
| | | Rev: GCCATCTTTCACAGCATCCT | | |
| | | Probe: ACCGAATGTGACAGCAATCTCGCA | | |
| *P. faecium* | 16S rRNA | Fwd: CCATCCTTTAGCGATAGCTTACT | 98 | This study |
| | | Rev: ACATTCCGAAAGGAGTGCTAATA | | |
| | | Probe: AGGCCATCTTTCTTCATCCTGCCA | | |
| *Phascolarctobacterium* sp. 377 | 16S rRNA | Fwd: GTAGGCAACCTGCCCTTTAG | 127 | This study |
| | | Rev: CCATCCTTTAGCGATAGCTTACAT | | |
| | | Probe: ATGTGACGCTCCTATCGCATGAGG | | |
| Total bacterial DNA | 16S rRNA | Fwd: AATAAATCATAAACTCCTACGGGAGGCAGCAGT | 204 | Brukner et al. (44) |
| | | Rev: AATAAATCATAACCTAGCTATTACCGCGGCTGCT | | |
| | | Probe: CGGCTAACTMCGTGCCAG | | |

included 12 samples with reads from *P. succinatutens*, four samples from *P. faecium*, four samples from *P.* sp 377, and four samples without *Phascolarctobacterium* reads. The total bacterial DNA load in each sample was estimated using the universal 16S rRNA as a target. The primer and probe sequences and the cycling conditions for the universal 16S rRNA gene amplification have been described elsewhere (44). qPCR data were analyzed with the ΔCt method ($\Delta Ct = Ct_{Target} - Ct_{Total\ DNA}$) using the 16S rRNA gene as a reference. Relative abundance was calculated by $2^{-\Delta Ct}$.

## Genome analyses

### 16S rRNA gene

To compare ASVs across studies, we made a phylogenetic tree based on the V4 region from NORCCAP and reference genomes for the three *Phascolarctobacterium* species identified in CRCAhus (*P. succinatutens*, *P. faecium*, and *P.* sp.377). Initially, we created a BLAST database of the 16S V4 region of the CRCAhus ASVs using the makeblastdb (v2.13.0) command from BLAST+ NCBI toolbox with default settings (45). Blastn was then employed to extract the corresponding V4 region from the NORCCAP and reference sequences (*P. succinatutens* (GCA 017851075.1), *P.* sp 377 (AB739694.1), and *P. faecium* (AP025563.1)). The V4 sequences from CRCAhus, NORCCAP, and reference genomes were then aligned by Multiple Alignment using Fast Fourier Transform (MAFFT, v7) (46). A maximum likelihood phylogenetic tree was constructed with IQ-TREE (47) (v2.2) using F18 +F substitution model and bootstrapping set to 1,000. The resulting tree was visualized using the Interactive Tree of Life (iTOL, v6) (48). ASVs with over 97% similarity to a reference sequence were collapsed into one ASV for all subsequent analyses.

### Metagenome data

All CRCbiome genomes belonging to the *Phascolarctobacterium* genus were annotated using Dram (49) (v1.4) with default settings using the databases KOfam (50) (accessed 31.10.2022), dbCAN (51) (accessed 08.09.2022), and Uniref90 (52) (accessed 14.11.2022). Identified protein-coding gene sequences were then used as input for a pangenome analysis using Roary (53) (v3.13), based on the identification of gene clusters with a 70% identity cutoff for protein similarity. Gene clusters within the species-specific core were defined as those found in 95% of the genomes from one species and in 0% of the other two. Genus-level core was defined as those genes present in ≥95% of genomes regardless of species. All genomes were aligned using MAFFT (54) (v7.520) and a tree was constructed using IQ-TREE (v2.2) with GTR + F + R7 substitution model and visualized using ITOL (v6). Pairwise average nucleotide identity (ANI) between the genomes was calculated based on tetranucleotide frequencies using the Python package pyani (v0.2.12).

## Statistics

Associations between *Phascolarctobacterium* species abundance and participant characteristics were evaluated in separate linear models for each species and variable. Abundance was coded as the dependent variable and participant characteristics as independent variables, adjusting for sex, age, screening center (Telemark or Oslo for NORCCAP, and Moss or Bærum for CRCbiome), or study of origin (total seven studies for curatedMG) to avoid confounding, as these variables are known to be associated with both microbial composition and other participant characteristics. Here, relative abundances were log-transformed, with 0 replaced by a pseudo count, defined as half the lowest observed relative abundance of the feature. The participant characteristics evaluated included clinical group (CRC, adenoma, or controls), lifestyle, and dietary factors. Diet variables included were energy intake (kcal/day), macronutrients (in energy percentage (E%)), and alcohol and fiber (in g/day) as described in reference (55). Lifestyle and demographic variables included were national background, education, occupation, marital status, body mass index, physical activity level, use of antibiotics and antacids in the past 3 months, smoking and snus habits, and the healthy lifestyle index (further details in Kværner et al. (23) and Istvan et al. (55)). The relationship between *Phascolarctobacterium* species relative abundance and FIT values was assessed using an ordinal logistic regression model adjusted for sex and age, implemented with the function polr from the R package MASS(56) (v7.3–60). Here, FIT values (in μg hemoglobin/g feces) were categorized into four groups based on their level of hemoglobin (group 1 = 15–20, group 2 = 20–35, group 3 = 35–70, group 4 = >70). Group differences in the prevalence of *Phascolarctobacterium* species were evaluated using a chi-squared test.

To assess the correlation between the relative abundance of *Phascolarctobacterium* species as estimated using NGS and qPCR, we performed Spearman correlation analysis. Pairwise co-occurrence of *Phascolarctobacterium* species was quantified as a percentage, calculated by dividing the number of sample pairs featuring two species by the total sample count within the data set and multiplying by 100. To evaluate whether the dominant *Phascolarctobacterium* species were associated with distinct microbial communities, a permutational multivariate analysis of variance (PERMANOVA) test was conducted using the adonis2 function from the vegan package(57) (v.2.5–7) based on Bray-Curtis distances of relative species abundance. Here, participants were categorized according to *Phascolarctobacterium* presence: those with reads exclusively mapping to *P. succinatutens*, *P.* sp 377, or *P. faecium*; those with reads mapping to two or more species; and those with no *Phascolarctobacterium* reads. The PERMANOVA test was adjusted for sex, age, and screening center. Cor_test from the package rstatix (58) (v.0.7.0) was used to calculate Spearman's correlation between the relative abundance of the three *Phascolarctobacterium* species and all other species or virus OTUs (vOTUs) (55). Before species-correlation analysis, a 5% prevalence filtration was performed. Correlation networks were visualized using Cytoscape (59) (v3.9.0).

Utilizing the results from the pangenome analyses, a chi-squared or Fisher's exact test was used to identify significant deviations in the prevalence of carbohydrate-active enzymes (CAZY) and Kyoto Encyclopedia of Genes and Genomes (KEGG) genes across CRC, adenoma, and control groups. In addition, we compared the prevalence of CAZy and KEGG genes within each *Phascolarctobacterium* species against the other two species combined. The KEGG genes with varying distribution across species were used for the pathway overrepresentation analysis with MicrobiomeProfiler (60) (v1.4.0).

All statistical analyses were performed using the R software (v4.1.0), with the main package being tidyverse (61) (v.1.3.1). Nominal statistical significance was considered for $P < 0.05$. Adjustment for multiple testing was performed using the Benjamini-Hochberg false discovery rate (FDR) (62), with FDR < 0.05 being considered statistically significant. Code available on https://github.com/Rounge-lab/Phascolarctobacterium_CRC.

## RESULTS

### Participant characteristics

In total, data from 2350 participants from three Norwegian CRC-related cohorts and the international collection of data sets available as curatedMG were analyzed (Table 2). The distribution of men and women was similar across data sets, with the percentage of women ranging from 39 to 44%.

### Phylogenetic comparison of the *Phascolarctobacterium* ASVs and reference genomes

We assessed the phylogenetic relationship between *Phascolarctobacterium* ASVs, including two CRC-associated ASVs and *Phascolarctobacterium* reference genomes. The CRC-associated ASVs identified in the NORCCAP data set (5) clustered with the *P. succinatutens* 16S rRNA gene reference sequence, whereas the CRC-associated ASVs identified in the CRCAhus data set (6) clustered with the 16S rRNA gene sequences from *P.* sp 377 (Fig. 2A). *Phascolarctobacterium* ASVs from paired biopsy and fecal samples (CRCAhus) clustered exclusively together. The CRCAhus ASVs clustered with *P. succinatutens* (6 ASVs), *P.* sp 377 (4 ASVs), and *P. faecium* (3 ASVs) reference genomes. In NORCCAP stool samples, 37, 1, and 2 ASVs clustered with *P. succinatutens, P.* sp 377, and *P. faecium* references, respectively. These results show that the ASVs represent three distinct species of *Phascolarctobacterium* and that the CRC-associated ASVs represent two independent *Phascolarctobacterium* species.

### qPCR confirms phylogenetically distinct ASVs and a CRC-association for *Phascolarctobacterium spp.*

To validate the phylogenetic discordance between *Phascolarctobacterium* ASVs identified in CRCAhus and NORCCAP, we established qPCR assays for *P. succinatutens, P.* sp 377, and *P. faecium*. Analytical specificity assessed for a panel of 50 bacterial species revealed all three assays to exhibit 100% specificity for each targeted *Phascolarctobacterium* species (Table S1). LOD for *P. succinatutens* and *P. faecium* assays was 1 fg/µL. qPCR assay detection rates for samples with sequencing reads for *P. succinatutens, P. sp* 377, and *P. faecium* were 96%, 94%, and 100%, respectively. The qPCR additionally detected (presence) of 4, 3, and 11 of *P. succinatutens, P.* sp 377, and *P. faecium*, respectively, where 3, 1, and 9 samples (seven from NORCCAP) had low abundance (Ct >32). With regards to metagenome data from NORCCAP and CRCbiome, qPCR analysis also confirmed the presence of the three species in these samples (Table S2). qPCR only detected five additional samples with either *Phascolarctobacterium* that did not have sequencing reads in the long-term stored NORCCAP samples. There was high concordance (100%)

**TABLE 2** Participant characteristics

| | CRCAhus[b] | NORCCAP 16S | NORCCAP MG[c] | CRCbiome | CuratedMG |
| --- | --- | --- | --- | --- | --- |
| | *n* = 72 | *n* = 143 | *n* = 46 | *n* = 1034 | *n* = 1055 |
| CRC, n (%) | 25 (35) | 28 (20) | 7 (15) | 66 (6) | 447 (42) |
| Adenoma, n (%) | 25 (35) | 62 (43) | 17 (37) | 298 (29) | 167 (16) |
| Control, n (%) | 22 (30) | 53 (37) | 22 (48) | 670 (65) | 441 (42) |
| Male, n (%) | 42 (58) | 86 (60) | 28 (61) | 582 (56) | 622 (59) |
| Female, n (%) | 30 (42) | 57 (40) | 18 (39) | 452 (44) | 433 (41) |
| Age, median (range) | 67.5 (30-87) | 57 (51–65) | 58,5 (53-65) | 67 (55–77) | 64 (21–88) |
| | | *16S* | | *Metagenome* | |
| *P. succinatutens,* n (%)[a] | 13 (16) | 37 (26) | 11 (24) | 156 (15) | 315 (30) |
| *P.* sp 377, n (%)[a] | 9 (13) | 1 (0.7) | 1 (2) | 88 (8) | 68 (6) |
| *P. faecium,* n (%)[a] | 22 (31) | 17 (12) | 4 (9) | 335 (32) | 320 (30) |

[a]from NGS relative abundance.
[b]CRCAhus included 72 participants with both stool (70) and biopsy samples.
[c]NORCCAP samples subset with metagenome sequencing.

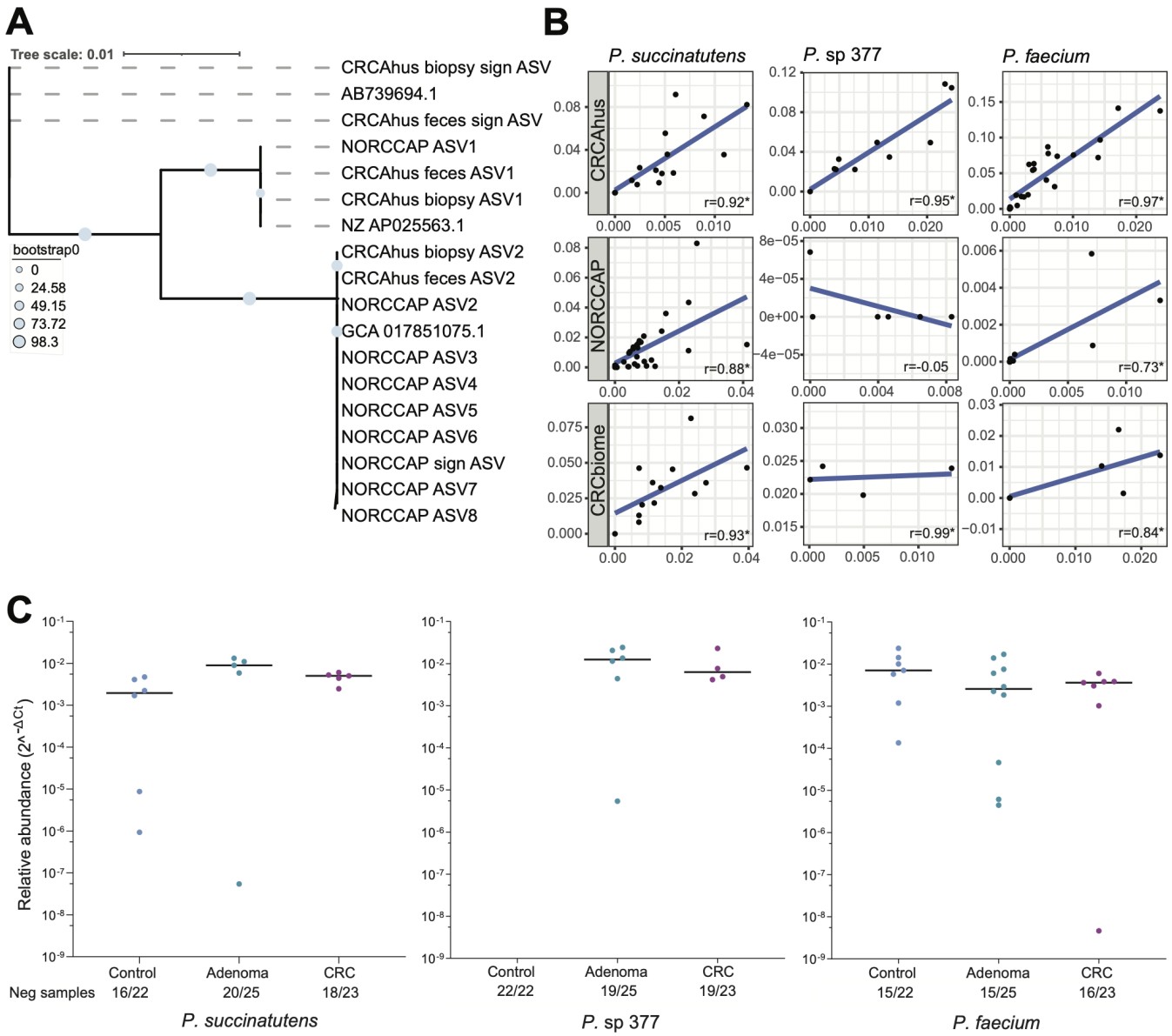

**FIG 2** (A) Phylogenetic tree showing that ASVs from CRCAhus and NORCCAP cluster with reference genomes for *P. succinatutens* (GCA 017851075.1), *Phascolarctobacterium sp*. (AB739694.1), and *P. faecium* (AP025563.1). The CRC-associated ASV from CRCAhus cluster in proximity to *P. sp 377* and the CRC-associated ASV from NORCCAP cluster in proximity to *P. succinatutens*. (B) Scatter plot illustrating the relationship between relative abundance from NGS data on the y-axis and relative abundance from qPCR on the x-axis. Each point represents one sample. Data are presented for *P. succinatutens*, *P. sp 377*, and *P. faecium* per data set (CRCAhus feces, NORCCAP, and CRCbiome). NORCCAP and CRCbiome samples were selected based on relative abundance data. (C) Relative abundance of *Phascolarctobacterium* spp in fecal samples from CRCAhus. While *P. succinatutens* and *P. faecium* were present in all three groups, the uncultured *P. sp 377* was not found in the control group. Each point represents one sample. The number of negative samples with 0 abundance is indicated on the x-axis (neg samples). *P*-value <0.001. r = Spearman's correlation coefficient.

between *Phascolarctobacterium* relative abundance detection in CRCbiome FIT samples and qPCR. Overall, this indicates high qPCR sensitivity across sample types and storage conditions.

Our results showed a high concordance between relative abundance from 16S rRNA gene sequencing, shotgun metagenome sequencing, and qPCR. Spearman's correlation coefficients of 0.92, 0.95, and 0.97 for *P. succinatutens*, *P. sp 377*, and *P. faecium*, respectively (all *P* < 0.01, Fig. 2B; Table S2), was observed in CRCAhus. In NORCCAP, 16 S *P. succinatutens* and *P. faecium* showed a significant positive Spearman correlation (0.88,

0.73, $P < 0.05$), but *P. sp* 377 did not (−0.05, $P = 0.7$). In the NORCCAP MG and CRCbiome cohorts, *P. succinatutens, P.* sp 377, and *P. faecium* showed positive Spearman correlations (0.97, 0.99, and 0.73 for NORCCAP MG, all $P < 0.01$, and 0.93, 0.99, and 0.84 for CRCbiome, all $P < 0.01$). In accordance with 16S rRNA sequencing-based detection in the CRCAhus study, qPCR results identified *P.* sp 377 in 6/25 adenomas and 4/23 CRC cases but were absent from the control group (Fig. 2C).

## CuratedMG metagenomes confirm the association between CRC and *P. succinatutens*

We further investigated the association between adenoma/CRC cases and abundance of the three *Phascolarctobacterium* species in two large and independent CRC-related data sets, namely CRCbiome and curatedMG. The results showed a positive association between *P. succinatutens* and adenomas/CRC in curatedMG (Fig. 3A; Table S3, linear mixed-effects models [lme]—see methods; all $P < 0.05$). *P.* sp 377 was not associated with adenomas or CRC in either data set. *P. faecium* was negatively associated with adenomas in curatedMG (lme, $P = 0.014$).

## Sex specificity of *P. faecium* and *P. succinatutens*

We also observed an association between *Phascolarctobacterium* species and sex. Men exhibited a higher abundance of *P. succinatutens* in CRCbiome and curatedMG data sets (Fig. 3A; Table S3, lme, $P < 0.05$), while women showed a higher abundance of *P. faecium* in curatedMG. Subsequent presence/absence analysis confirmed a higher presence of *P. succinatutens* in men across NORCCAP MG, CRCbiome, and curatedMG, and a greater prevalence of *P. faecium* in women in curatedMG (Fig. 3B; Table S4, chi-squared test, all $P < 0.05$).

## *Phascolarctobacterium* species are mutually exclusive and have distinct microbial partners

We further investigated the characteristics of the microbiome, the prevalence of *Phascolarctobacterium* species in participants' microbiomes, and their interactions with other microbes. We explored the extent of *Phascolarctobacterium* species co-occurrence across samples and study populations. We found a low rate of co-occurrence between the *Phascolarctobacterium* species in all data sets (Fig. 4A; Table S5). The highest pairwise co-occurrence was observed in CRCAhus biopsy samples between *P. faecium* and *P.* sp 377 (6%). For all other data sets, co-occurrence was less than 3% and no samples had all three species across data sets. There was also a significant compositional difference between samples with different dominating *Phascolarctobacterium* species in CRCbiome (PERMANOVA $P = 0.001$, $R^2 = 0.02$, Fig. 4B; Fig. S1) and curatedMG (PERMANOVA $P = 0.001$ and $R^2 = 0.02$).

For all data sets, we identified 321 species with significant correlation to one or more *Phascolarctobacterium* species where 248 showed a positive correlation. Forty-one species showed consistent correlations across metagenome data sets. *Dialister invisus* exhibited negative correlations with all three *Phascolarctobacterium* species (Fig. 4C; Table S6), suggesting that this species could also be mutually exclusive. On the other hand, *Bacteroides salyersiae* was positively correlated to both *P. faecium and P. succinatutens*. There were also five other *Bacteroides* species that showed a positive correlation to *Phascolarctobacterium* species. We have recently characterized viral diversity in CRCbiome samples (55). Here we detected 12 vOTUs with significant associations to one or more *Phascolarctobacterium* species (Fig. S2). In contrast to the predominantly positive associations observed between bacteria, 11 out of 12 significant associations for viruses were negative, and only one had a positive association with *P. faecium*, but not with other *Phascolarctobacterium*.

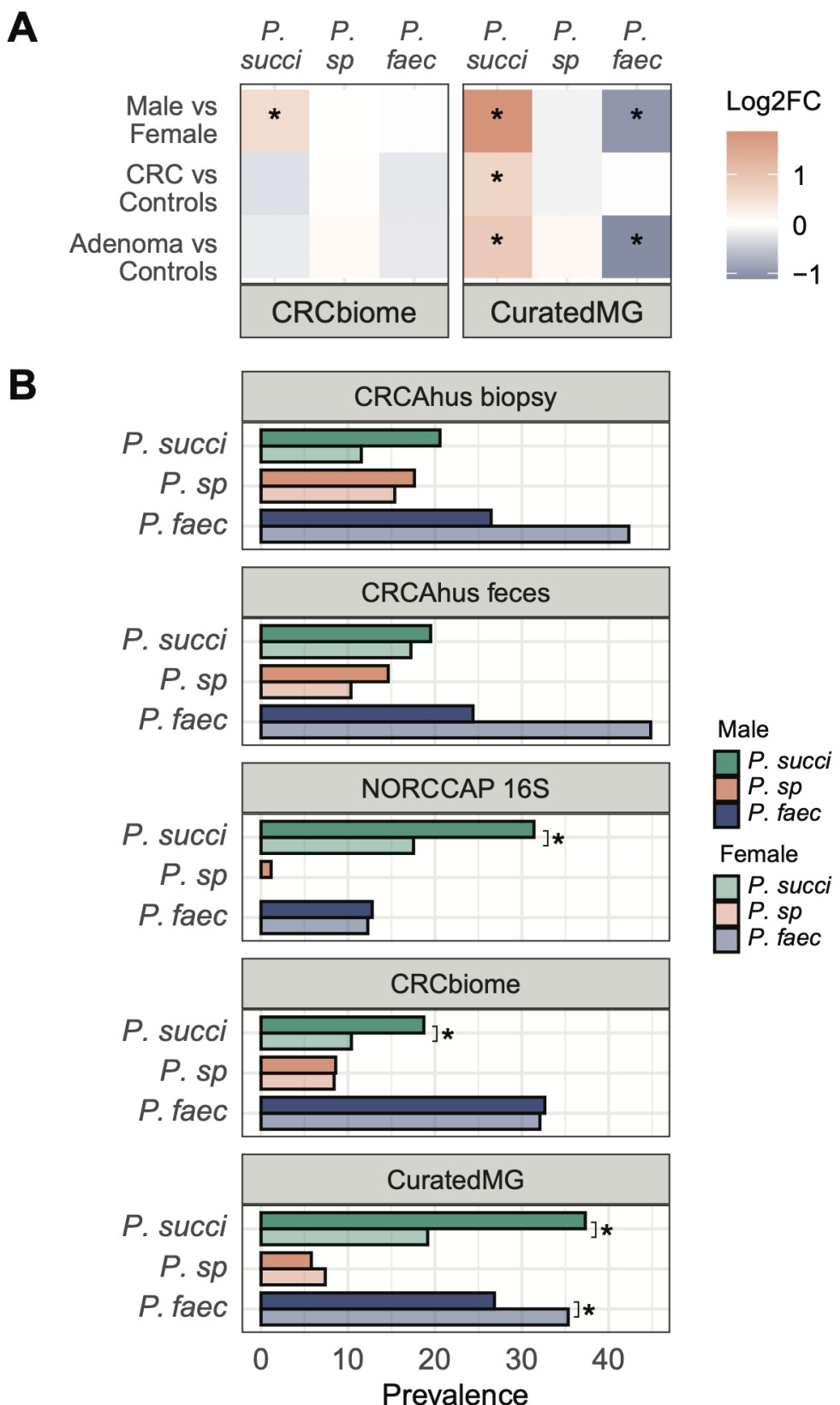

FIG 3 (A) Summary of multivariate linear models adjusting for sex, age and, for region (CRCbiome) and study (curatedMG). Color indicates log2 fold change, with red indicating a higher abundance and blue indicating a lower abundance compared to the reference group. A significantly increased abundance of *P. succinatutens* was observed in adenoma/CRC compared to

Fig 3 (Continued)

controls in curatedMG and a lower abundance of *P. faecium* in adenomas versus controls. Significantly higher abundance of *P. succinatutens* in males was observed in both CRCbiome and curatedMG. (B) Percentage of samples containing each of the three *Phascolarctobacterium* species, categorized by sex. Men displayed a higher prevalence of *P. succinatutens* compared to women in the NORCCAP, CRCbiome, and curatedMG data sets. *P. succi = P. succinatutens; P. sp = P. sp 377; P. faec = P. faecium,* \**P*-value <0.05 for lme (A) and chi-square test (B).

## Association of *Phascolarctobacterium* species abundance with education but not with diet and fecal blood concentration

*P. faecium* and *P. succinatutens* both use succinate as a primary carbon source; therefore, we investigated whether the relative abundance of *Phascolarctobacterium* species was associated with diet and other lifestyle factors. Here, we employed the CRCbiome data set with dietary and lifestyle information. After adjusting for sex, age, and screening center in linear regression models, there was a significant association with alcohol consumption and increased abundance of *P.* sp 377 ($P$ = 0.018 and $P$adj >0.05; Table S7). High school ($P$adj = 0.04) and university education ($P$ = 0.005 and $P$adj >0.05) were associated with lower abundance of *P. succinatutens*. University education ($P$ = 0.03 and $P$adj >0.05) and those not married or cohabiting ($P$ = 0.03 and $P$adj >0.05) was associated with higher and lower abundance of *P. faecium,* respectively. The concentration of blood in stool was not associated with the abundance of either of the three *Phascolarctobacterium* species (all $P$ > 0.05).

## Pangenome variability among *Phascolarctobacterium* species

Based on metagenome sequencing data from CRCbiome, 221 high-quality genomes of the *Phascolarctobacterium* genus were identified. Fifty-two genomes were annotated as *P. succinatutens*, 131 as *P. faecium*, and 32 as *P.* sp 377, and their phylogenetic topology (Fig. 5A), that is, relatedness between species, corresponded with the 16S-based tree (Fig. 2A). Mean within-species ANI was 99.9%, 99.9%, and 99.8% for *P. faecium*, *P.* sp 377, and *P. succinatutens,* respectively, and the mean between-species ANI was 73.9% (Fig. 5B).

Pangenome analysis for all *Phascolarctobacterium* genomes identified 25,847 gene clusters, with 1,423 of them being ubiquitous (≥95%) within a species, and not found in the others (species-specific cores). On average, each genome contained 2,065 gene clusters. Specifically, the average for *P. succinatutens* was 2,071, *P.* sp 377 1,752, and *P. faecium* 2,153 gene clusters. Only 197 gene clusters were identified in ≥95% of *Phascolarctobacterium* genomes (genus-level core). In all, 17,127 gene clusters were annotated with UniRef, 1,804 with KEGG pathways, and 65 with CAZy annotations. All species-specific cores had multidrug resistance genes, metallobetalactamases, 2-thiouracil desulfurase enabling $H_2S$ production, and contained various virulence factors. For example, *P. succinatutens* genomes contained amylovoran and holin-like protein genes (Table S8); *P.* sp 377—holin-like protein genes (Table S9); and *P. faecium*—heme-binding protein, exfoliative toxin, hemolysis, and immunity protein genes (Table S10).

There was an over-representation of genes within the porphyrin and chlorophyll metabolism KEGG pathway in *P. succinatutens*. Glyxoylate and diglyxolyate metabolism and glycine, serine, and threonine metabolism KEGG pathways were over-represented in *P.* sp 377. *P. faecium* genomes were enriched in histidine metabolism, ABC transporters, two-component system, phosphonate and phosphinate metabolism, and biosynthesis of amino acids KEGG pathway genes (chi-square test, all $P$adj < 0.05, Fig. 5C; Table S11).

With regard to carbohydrate-active enzymes, two CAZy entries belonging to the glycoside hydrolases family and one belonging to the carbohydrate binding molecules family were significantly more prevalent in *P. succinatutens* compared to the two other species. Three CAZy belonging to glycoside hydrolases and four belonging to glycosyl transferases family were more prevalent in *P. faecium* compared to the two others (chi-square test, all $P$adj < 0.05, Fig. 5D; Table S12). Glycoside hydrolase 171 was present

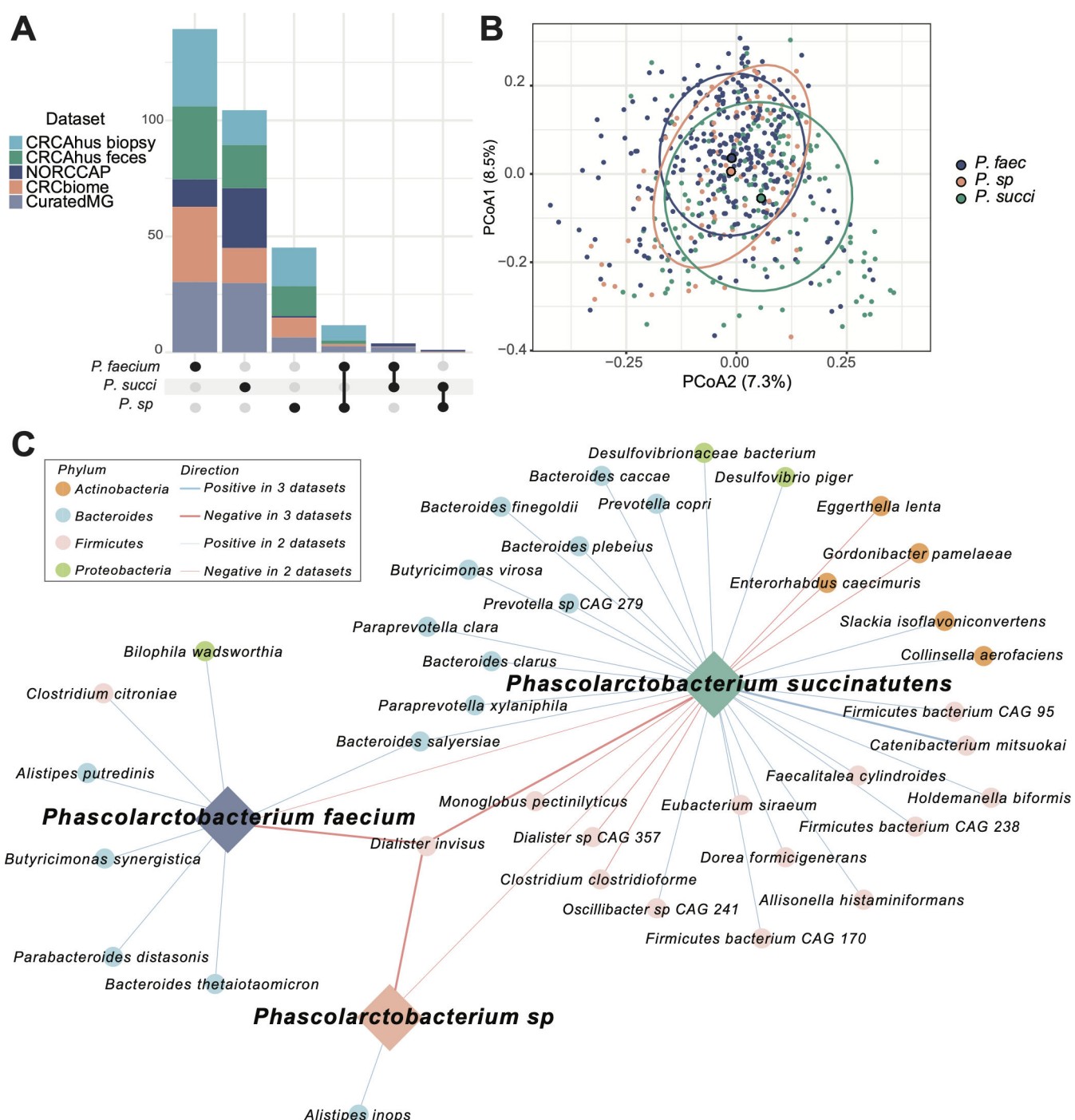

**FIG 4** (A) Upset plot illustrating the co-occurrence of *Phascolarctobacterium* species in all five datasets. No samples had all three species present across data sets. (B) PCoA plot showing the microbial composition for the CRCbiome samples, where the groups are defined based on the presence of one dominating *Phascolarctobacterium* species. PERMANOVA test showed a significant difference between the three groups (*P* = 0.001) with an R$^2$ of 0.02. (C) Correlation network plot of the 41 species with FDR significant, consistent correlations across at least two of the metagenome data sets (NORCCAP MG, CRCbiome, and curatedMG). Edge colors represent phyla. The red line color indicates negative correlations and blue indicates positive correlations. Line thickness indicates a number of data sets the correlation was observed in. *P. succi* = *P. succinatutens*; *P. sp* = *P. sp* 377; *P. faec* = *P. faecium*.

in all *P. succinatutens* and *P. faecium,* but completely missing in *P. sp*. Glycoside hydrolase 33 was exclusively found in *P. succinatutens* (88% of genomes) and glycoside hydrolase 3 was exclusively in *P. faecium* (98% of genomes).

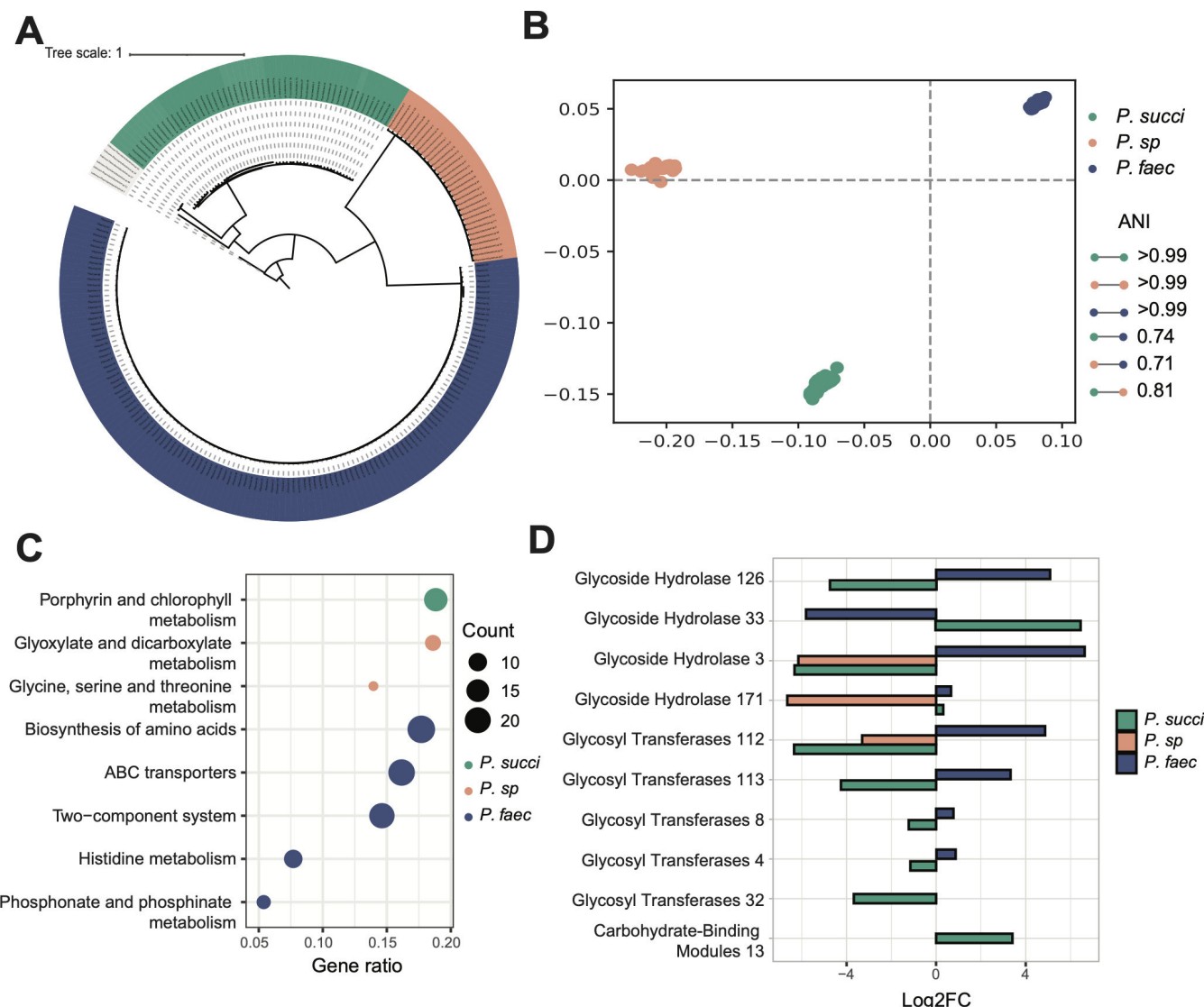

**FIG 5** *Phascolarctobacterium* species genome comparison. (A) Core-genome maximum likelihood tree representing all *Phascolarctobacterium* genomes in CRCbiome used for pangenome analyses, 52 genomes from *P. succinatutens*, 32 from *P. sp* 377, and 131 from *P. faecium*. (B) Multi-dimensional scaling of *Phascolarctobacterium* genomes based on their pairwise ANI distances. The average within species ANI and between species ANI are presented in the legend. (C) Enrichment analysis of pathways with a significant over-representation of KEGG genes from either *P. succinatutens*, *P. sp* 377, or *P. faecium*. KEGG genes included in the analyses were those that were significantly different between whichever species against the two others combined, as determined by a chi-square test (*P*adj <0.05). The size of the dot point represents the number of KEGG genes within the relevant pathway. (D) Log2FC of the significantly different CAZy enzymes between one species versus the two others combined as determined by a chi-square test (*P*adj <0.05). Only samples from CRCbiome are included in these analyses. ANI = Average Nucleotide Identity; Log2FC = Log2 fold-change; *P. succi = P. succinatutens*; *P. sp = P. sp* 377; *P. faec = P. faecium*.

## DISCUSSION

Based on our findings in two independent Norwegian cohorts, we replicate an association between the increased abundance of *P. succinatutens* and adenoma/CRC in the large international curatedMG data set. Three species were identified within the *Phascolarctobacterium* genus to be nearly mutually exclusive, forming distinct microbial communities, potentially defining a CRC-relevant microbial state. *P. succinatutens* was more common in men, in line with their increased CRC risk. Together, this puts *P. succinatutens* on the list of highly relevant and reproducibly CRC-associated bacteria.

In this study, we describe three distinct species within the genus *Phascolarctobacterium*. These were *P. succinatutens*, *P. faecium,* and one uncultured species referred to as *P.*

sp 377, all with a between-species ANI of <95% and a limited core genome. Using qPCR, we linked >200 high-quality genomes from the species mentioned encompassing four datasets to our previously identified CRC-associated 16S rRNA gene ASVs. The PCR assay was more sensitive than NGS in detecting low abundant *Phascolarctobacterium*.

We observed a mutually exclusive relationship between *Phascolarctobacterium* species across data sets and regardless of methods. The three different species of *Phascolarctobacterium* formed species-specific bacterial and viral networks, in addition to different overall community structures. *P. faecium* composition was more similar to those without any *Phascolarctobacterium*, whereas *P. succinatutens* was markedly distinct. These distinct community structures could indicate competition for resources or niche adaptation. Interestingly, all *Phascolarctobacterium* species were negatively correlated with *Dialister* and tended to have positive correlations with *Bacteroides*, suggesting that these community structures extend beyond the *Phascolarctobacterium* genus.

Bacteria in the large intestine ferment complex carbohydrates and fibers and produce short-chain fatty acids (SCFA), primarily acetate, butyrate, and propionate. SCFAs, and especially butyrate, have been proposed as potential biomarkers for CRC as they play a role in strengthening the gut barrier and modulation of immune responses (63). Succinate is an SCFA precursor and serves as a substrate for several bacteria, including *Phascolarctobacterium* and *Dialister* (64). This common reliance on succinate makes them potential competitors and might explain the observed negative correlations. The positive feedback loop between succinate-producing *Bacteroides thetaiotaomicron* and both *Dialister hominis* (65) and *P. faecium* (10) has been demonstrated.

The three *Phascolarctobacterium* species shared only a small conserved genus-level core genome of about 0.76% of their genes, supporting distinct niche adaptation. For example, we observed significant variations in metabolic capacity. Interestingly, glycoside hydrolase family 33 was found only in *P. succinatutens*. Glycoside hydrolase family 33 comprises sialidases that break down sialic acid from the diet (mainly red meat) and potentially from the mucus layer in the intestine (66) causing inflammation (67, 68). By contrast, Glycoside hydrolase family 3 was found exclusively, and in almost 100% of *P. faecium* genomes and is involved in a range of mechanisms including bacterial pathogen defense, cell-wall remodeling, energy metabolism, and cellulosic biomass degradation (69). Carbon starvation protein, a membrane protein, was found to be unique to *P.* sp 377. Carbon starvation is exhibited by bacteria when they experience a depletion of carbon sources for their metabolic process (70) and may provide *P. sp* 377 a selective advantage in nutrient-limited conditions.

Bacterial virulence factors are employed in bacterial warfare and are often detrimental to host health (71–73). We found different virulence factors for the three species. Holin-like protein was present in only *P. succinatutens* and *P.* sp 377. Holin-like proteins control cell wall lysis by producing pores in the cell membrane and can be involved in biofilm formation (74) contributing to chronic inflammation in the colon, a known risk factor for CRC (75, 76). Another gene involved in biofilm formation, TabA, was specific to *P. succinatutens*. We also found an overrepresentation of porphyrin and chlorophyll metabolism in *P. succinatutens*. Succinate is the main precursor and porphyrin is an intermediate of heme production, which is closely linked to the TCA cycle. Succinyl-CoA is the intermediate compound of succinate in the TCA cycle and is released upon production of an ATP molecule (77). In our previous work, we showed a lower abundance of several pathways related to heme biosynthesis in high-risk adenomas compared to healthy controls (5). Haem-binding uptake protein (Tiki superfamily) and hemolysin III protein were identified as distinct from *P. faecium*. Tiki proteins may function as Wnt proteases, counteracting the Wnt signaling pathway (78), a pathway which is commonly deregulated in CRC (79). Hemolysin III exhibits hemolytic activity and contributes to the destruction of erythrocytes by pore formation (80). Together, our findings from the pangenome analyses contribute to a deeper understanding of the functional diversity of *Phascolarctobacterium* species in the CRC microbiome.

We replicate our previous findings of an association between the increased abundance of *P. succinatutens* and adenomas/CRC. Several studies have reported similar associations at the genus level (13–15), with few having looked at the species level. Both our previous work including 17 years of follow-up (5), and Yachida et al. (12) found an increased abundance of *P. succinatutens* in the early stages of CRC. We observed a lower abundance of *P. faecium* in adenomas and also low levels of co-occurrences between *P. succinatutens* and *P. faecium*. This could indicate that the gut community might shift from a low-risk *P. faecium* community to a high-risk *P. succinatutens* community in early cancerogenesis.

Noteworthy, we found a higher prevalence of *P. succinatutens* in men than in women across cohorts independent of the colonoscopy outcome. Men have an elevated risk for CRC (81), often attributed to lifestyle and dietary factors (82, 83). We did, however, not find an association between *Phascolarctobacterium* abundance and host diet and lifestyle, nor with the presence of blood in the stool. On the contrary, the observed association with education could be a proxy for socioeconomic status where low socioeconomic status has been linked to an increased risk of CRC (84, 85).

Here we report consistent findings of *Phascolarctobacterium* across cohorts with different methods, which emphasizes the reliability of our results and strengthens the validity of the study. However, this study has some limitations. All participants in the CRCbiome study are FIT positive and therefore have blood in their stool something which has been suggested to alter the microbiome composition (86) and could also be a sign of colonic inflammation. It may also introduce selection bias in the cohort. This may provide a reason why we did not observe an association between *Phascolarctobacterium* abundance and adenoma/CRC in the CRCbiome cohort.

External factors like smoking, diet, and gut flora may influence different stages along the adenoma-carcinoma sequence of events leading to bowel cancer. The interplay between *Phascolarctobacterium* species revealed in this study adds further to this complexity revealing possible CRC-associated microbial networks and genomic characteristics.

## Conclusion

Our study reveals that three *Phascolarctobacterium* species form distinct microbial communities in the gut, each possessing different virulence factors and metabolic capabilities. We found that microbiome composition varies significantly according to which *Phascolarctobacterium* species is dominating. The verification of the *P. succinatutens* association with adenomas and CRC, and the observation of an increased abundance of *P. faecium* in controls, suggests that the gut community might shift from a low-risk *P. faecium* community to a high-risk *P. succinatutens* community in early cancerogenesis.

## ACKNOWLEDGMENTS

We would like to acknowledge Jan-Inge Nordby for his work on preparing both NORCCAP and CRCbiome samples, and for performing the DNA extractions. Elina Vinberg has also contributed to sample handling and project coordination in both NORCCAP and CRCbiome projects. Library preparation and sequencing of NORCCAP and CRCbiome samples were performed at the FIMM Technology Centre supported by HiLIFE and Biocenter Finland. Therefore, we would like to thank Tiina Hannunen, Harri A. Kangas, and Pekka J. Ellonen for their service and good cooperation. We would also like to thank the members of our research groups Maja Sigerseth Jacobsen, Ane Sørlie Kværner, Paula Berstad, and Paula Istvan. Thank you for the great working environment and fruitful discussions. We thank the Department of Multidisciplinary Laboratory Science and Medical Biochemistry at Akershus University Hospital for providing laboratory facilities. We are grateful to Tone M. Tannæs, Aina E.F. Moen, Gro Gundersen, Eva Smedsrud, and John Christopher Noone for their contribution to sample extraction and sequencing.

This work was supported by the South-Eastern Norway Regional Health Authority under Grant numbers 2020056 and 2022067; Oslo Metropolitan University under Grant number 202401; and Akershus University Hospital. The CRCbiome study was supported by the Norwegian Cancer Society under Grant numbers 190179 and 198048. Sequencing of the NORCCAP samples was funded by the Cancer Registry of Norway funds.

T.B.R. and H.T. designed the research. C.B.J., E.B., E.A., T.B.R., T.S., H.T., and V.B. conducted the research. C.B.J., E.B., E.A., and T.S. analyzed data or performed statistical analysis. C.B.J. and T.S. drafted the paper. All authors read and approved the final manuscript.

## AUTHOR AFFILIATIONS

[1]Department of Research, Cancer Registry of Norway, Norwegian Institute of Public Health, Oslo, Norway
[2]Department of Tumor Biology, Oslo University Hospital, Oslo, Norway
[3]Center for Bioinformatics, Department of Pharmacy, University of Oslo, Oslo, Norway
[4]Department of Life Sciences and Health, Oslo Metropolitan University, Oslo, Norway
[5]Center for Bioinformatics, Department of Informatics, University of Oslo, Oslo, Norway
[6]Section for Colorectal Cancer Screening, Cancer Registry of Norway, Norwegian Institute of Public Health, Oslo, Norway
[7]Telemark Hospital, Skien, Norway
[8]Department of Pathology, Akershus University Hospital, Lørenskog, Norway

## AUTHOR ORCIDs

Thulasika Senthakumaran (iD) http://orcid.org/0000-0003-0469-6327
Einar Birkeland (iD) http://orcid.org/0000-0002-9361-7987
Trine B. Rounge (iD) http://orcid.org/0000-0003-2677-2722

## FUNDING

| Funder | Grant(s) | Author(s) |
| --- | --- | --- |
| Ministry of Health and Care Services \| Helse Sør-Øst RHF (sorost) | 2020056 | Trine B. Rounge |
| Ministry of Health and Care Services \| Helse Sør-Øst RHF (sorost) | 2022067 | Trine B. Rounge |
| Kreftforeningen (NCS) | 190179 | Trine B. Rounge |
| Kreftforeningen (NCS) | 198048 | Trine B. Rounge |
| Oslo Metropolitan University | 202401 | Hege Tunsjø |

## DATA AVAILABILITY

Data from the CRCbiome project have been deposited in the database Federated EGA under accession code EGAS50000000170 and the curated Metagenomic data are available here: https://waldronlab.io/curatedMetagenomicData/index.html. Due to the sensitive nature of the data derived from human subjects, including personal health information, analyses and sharing of data from cohorts in this project must comply with the General Data Protection Regulation (GDPR). How to get access to the data is described here: https://www.mn.uio.no/sbi/english/groups/rounge-group/crcbiome/. The custom R scripts used in this study are available at: https://github.com/Rounge-lab/Phascolarctobacterium_CRC.

## ETHICS APPROVAL

The CRCAhus study, BCSN trial, the CRCbiome study, and the NORCCAP trial have been approved by the Regional Committee for Medical and Health-related Research Ethics in

Southeast Norway (REK ref: 2012/1944, 2011/1272, 63148, and 22337, respectively). The CRCAhus study also received approval from the data protection manager at Akershus University Hospital. The BCSN trial is registered at clinicaltrials.gov (Clinical Trial (NCT) no.: 01538550).

## ADDITIONAL FILES

The following material is available online.

### Supplemental Material

**Supplemental figures (mSystems00734-24-s0001.pdf).** Figures S1 and S2.
**Supplemental tables (mSystems00734-24-s0002.xlsx).** Tables S1 to S12.

### Open Peer Review

**PEER REVIEW HISTORY (review-history.pdf).** An accounting of the reviewer comments and feedback.

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
