## [Reviewer comments · mSystems]

Species-level verification of *Phascolarctobacterium* association to colorectal cancer

Cecilie Bucher-Johannessen, Thulasika Senthakumaran, ekaterina avershina, Einar Birkeland, Geir Hoff, Vahid Bemanian, Hege Tunsjø, and Trine Rounge

Corresponding Author(s): Trine Rounge, Universitetet i Oslo

Review Timeline:

Submission Date:	May 28, 2024
Editorial Decision:	July 14, 2024
Revision Received:	August 19, 2024
Accepted:	August 24, 2024

Editor: Jotham Suez

Reviewer(s): The reviewers have opted to remain anonymous.

Transaction Report:

DOI: <https://doi.org/10.1128/msystems.00734-24>

Re: mSystems00734-24 (Species-level verification of *Phascolarctobacterium* association to colorectal cancer)

Dear Prof. Rounge:

Thank you for the privilege of reviewing your work. It has now been reviewed by two experts in the field. As you will see below, they were enthusiastic about the work and have requested minor modifications. Their comments, as well as instructions from the mSystems editorial office, can be found below.

Revision Guidelines

Sincerely,
Jotham Suez
Editor
mSystems

Reviewer #1 (Comments for the Author):

I enjoyed reading the manuscript. It is well-structured and written clearly.

The manuscript is a continuation of the group's work on the association of bacterial species and colorectal cancer (CRC). Specifically, the team found *Phascolarctobacterium* correlated with CRC and adenomas in their preceding study. The aim of this paper is to strengthen the previous claims by including more cohorts (3 different Norwegian cohorts + data from published studies) and other methods (16S seq, metagenomics, qPCR) and the manuscript delivers that.

The value of the manuscript is solid evidence supporting the claims. It gives a higher level of confidence that the described association is a real signal rather than a spurious result. Due to that, even though the novelty of the paper is limited, I still believe it is valuable. Moreover, the paper provides an initial characterization of *Phascolarctobacterium* species associated with the disease.

Methods are well described; sequencing data and code are made available.

Comments:

Phylogenetic analysis (Figure 2A) was done on 16S gene. The whole genome data are available in the study, so it would be more meaningful to include such analysis at the whole genome level as well. That would further validate that the identified ASVs belong to *Phascolarctobacterium*.

Figure labels - several labels are tiny. I suggest you increase the size of the labels to ensure everyone can read them without magnifying the image.

"The CRC-associated ASVs from NORCCAP(5) and CRCAhus(6) studies clustered in proximity to 16S rRNA gene sequences from *P. succinatutens* and *P. sp 377* genomes, respectively (Figure 2A)." - the phrase 'clustered in proximity' is not specific. I think it needs rephrasing.

Reviewer #2 (Comments for the Author):

I read with interest the manuscript by Bucher-Johannessen et al., where they conduct a follow-up study to elucidate the associations of *Phascolarctobacterium* species in patients with CRC and adenomas. The analysis of the *Phascolarctobacterium* genus at the species level elucidates its potential differential ecological niche associated with dysbiosis in the gut microbiome, such as the case of *P. succinatutens* being more abundant in adenoma and CRC. The authors have conducted thorough work, not only identifying associations at the species level in their cohort but also utilizing the valuable resource `curatedMetagenomicData` to validate their findings against previous studies on CRC.

They follow a combined approach to identify the *Phascolarctobacterium* species by combining qPCR and sequencing data to further combine this new data with the type of colonic lesions, gender, among other variables. Microbial genera do not contain species that are functionally homogenous, and this study is an example based on the pangenome analysis. That being said, I have some minor comments to improve the clarity of the manuscript:

Can the authors comment on the logic behind adjusting their test for sex, age, and screening center?

Can the authors comment about BMI distribution and *Phascolarctobacterium* species?

Additionally:

In line 119, can the authors define what 'CT' means?

For the statistical test results, can the authors add the name of the test together with the adjusted p-values?

Point-by-point response “Species-level verification of *Phascolarctobacterium* association to colorectal cancer” by Bucher-Johannessen et al.

We would like to thank the reviewers for their enthusiasm and thorough revision of our manuscript. In this point-by-point response, we address the suggestions raised by the reviewers.

Reviewer #1:

Thank you for enjoying our manuscript. We are pleased you found it well-structured and written clearly.

Comments:

1. Phylogenetic analysis (Figure 2A) was done on 16S gene. The whole genome data are available in the study, so it would be more meaningful to include such analysis at the whole genome level as well. That would further validate that the identified ASVs belong to *Phascolarctobacterium*.

*Response: We agree that phylogenetic analyses at the genome level is more valuable than using the 16S rRNA gene sequences alone, even though we consider the 16S tree to provide strong evidence for the *Phascolarctobacterium* designation. In line with the suggestion of the reviewer, we have already included a phylogenetic analysis of *Phascolarctobacterium* genomes (presented in Figure 5A). The 16S rRNA gene tree in 2A and the core genome-based tree in 5A show corresponding structures. We have now included this observation to the manuscript (line 512-513). Further supporting the accurate designation of ASVs as *Phascolarctobacterium*, the species-specific qPCR assays confirmed the presence of the three *Phascolarctobacterium* species across datasets.*

2. Figure labels - several labels are tiny. I suggest you increase the size of the labels to ensure everyone can read them without magnifying the image.

Response: We have reviewed all the labels and made sure they comply with the journal minimum requirements. Although the genome labels in Figure 5A are too small to be readily legible, they are clearly color-coded on a species level to aid interpretability.

3. "The CRC-associated ASVs from NORCCAP(5) and CRCAhus(6) studies clustered in proximity to 16S rRNA gene sequences from *P. succinatutens* and *P. sp 377* genomes, respectively (Figure 2A)." - the phrase 'clustered in proximity' is not specific. I think it needs rephrasing.

*Response: We agree that this phrasing was unclear. Accordingly, we have rephrased this statement (lines 379-380): “The CRC-associated ASVs identified in the NORCCAP dataset clustered with the *P. succinatutens* 16S rRNA gene reference sequence, whereas the CRC-associated ASVs identified in the CRCAhus dataset clustered with the 16S rRNA gene sequences from *P. sp 377* genome (Figure 2A).”*

Reviewer #2

Thank you for finding our manuscript interesting.

1. Can the authors comment on the logic behind adjusting their test for sex, age, and screening center?

Response: To account for potential confounding, we included sex, age, and screening center as covariates in analyses of associations between microbial abundance and participant characteristics. These variables were chosen as they capture major sources of participant variation (screening center being a measure of geography), in particular the detection of colorectal lesions, and are also associated with microbial composition. While not comprehensive, these adjustments should alleviate confounding from these likely sources of bias. We have included a sentence on this in the manuscript (line 304-305).

2. Can the authors comment about BMI distribution and Phascolarctobacterium species?

Response: Data on BMI was available for the CRCbiome study population (Median = 26.5, IQR = 5.2), and was similar to the general Norwegian population (<https://www.fhi.no/globalassets/dokumenterfiler/rapporter/2021/rapport-nhus-2020.pdf>). While summary statistics for BMI is not reported in the manuscript, data on the BMI distribution for this population has been presented in Kværner et al 2018, Cancer Medicine). BMI did not have a statistically significant association with Phascolarctobacterium species abundance (see table S7 for details).

3. Additionally:

*Response: In line 119, can the authors define what 'CT' means?
The abbreviation has now been replaced with "computed tomography" (line 119).
For the statistical test results, can the authors add the name of the test together with the adjusted p-values?*

Line 410: added "Spearman"

Line 412: added "Spearman".

Line 435: added "linear mixed effects models – see methods"

Line 437: added "lme"

Line 442: added "lme"

Line 445: added "chi squared test"

Figure 3: added "for lme (A) and chi-square test (B)."

Line 470: added "PERMANOVA"

Line 471: added "PERMANOVA"

Line 501: added "linear regression models"

Line 534: added "chi-square test"

Line 540: added "chi-square test"

Re: mSystems00734-24R1 (Species-level verification of *Phascolarctobacterium* association to colorectal cancer)

Dear Prof. Trine B Rounge:

Thank you for addressing the reviewers' comments in your revised manuscript. I am happy to update you that your manuscript has been accepted, and I am forwarding it to the ASM production staff for publication. Your paper will first be checked to make sure all elements meet the technical requirements. ASM staff will contact you if anything needs to be revised before copyediting and production can begin. Otherwise, you will be notified when your proofs are ready to be viewed.

Sincerely,

Dr. Jotham Suez
Editor
mSystems